# Patient satisfaction in treatment of non-complex fractures and dislocations in general practice in the Netherlands: prospective cohort study protocol

Tjitte Verbeek,[1] Hans Arentsen,[2] Evert Jan Breet,[3] Machiel M Kuipers,[4] Pieter H W Lubbert,[5] Huibert Burger[1]

For numbered affiliations see end of article.

**Correspondence to**
Dr Tjitte Verbeek;
t.verbeek@umcg.nl

## ABSTRACT

**Introduction** Diagnosis and treatment of fractures and dislocations are mostly performed in hospital settings. However, equal care for patients with non-complex fractures or dislocations ('minor trauma care') may be provided in general practice. While substitution of care from secondary to primary care settings is stimulated by governments and insurers, it is unknown what the effects are on patient satisfaction level. Therefore, our primary objective is to determine the effect of minor trauma care delivered in a general practice as compared with a hospital on patient satisfaction. Secondary objectives are to assess the effects on treatment outcomes, cost-effectiveness and time consumption.

**Methods and analysis** In a prospective cohort study, we will include 200 patients aged 12 and over with an X-ray confirmed diagnosis of a non-complex fracture or dislocation out of whom 100 treated in a general practice and 100 in a secondary care hospital, both located in the Netherlands. All treatment procedures and follow-up will be done in accordance to the hospital's standards of trauma care. Study assessments will be performed pre-treatment, and 1, 6 and 12 weeks after treatment. Data collected include demographics, patient satisfaction and patient-reported outcomes including physical functioning, complications, pain scores and treatment-related costs. The primary outcome patient satisfaction measured at 12 weeks will be compared between the settings and additionally multivariable regression will be performed to assess potential confounding effects of unbalanced prognostic factors. Treatment outcomes and time consumption will be analysed following the same approach while cost-effectiveness will be assessed using an incremental cost-effectiveness ratio. Subsequently, results will be discussed using focus groups consisting of patients (n=15) and healthcare providers.

**Ethics and dissemination** The Medical Ethics Committee from the University Medical Center Groningen reviewed this study protocol and granted exemption from ethical approval (METc UMCG 2017/277). Study results will be presented at (inter)national conferences and published in peer-reviewed journals.

**Trial registration number** NCT03506958; Pre-results.

## Strengths and limitations of this study

► The observational cohort study design provides generalisable insights about trauma care, provided in general practice.
► Local two-arm setting provides a clear comparison of trauma care in general practice with hospital trauma care.
► Broad inclusion criteria are used to obtain a representative sample of the study population.
► Absence of randomisation might lead to bias due to the influence of uncontrolled or unbalanced variables or due to possible differences among referring general practitioners.
► Possible bias as a result of loss of follow-up or patients unwilling to complete questionnaires.

## INTRODUCTION

In the Netherlands, diagnostics and treatment of bone fractures and dislocations are mostly organised in the secondary care setting. When a fracture or dislocation is presumed, most general practitioners refer the patient to an X-ray facility in a nearby hospital. When the fracture or dislocation is X-ray confirmed, an emergency care doctor or trauma surgeon generally provides the treatment and follow-up. In contrast, since 2017 a unique general practice in the Netherlands provides equal care for patients with non-complex fractures or dislocations.[1] In this practice regular X-ray diagnostics are used, which are digitally transmitted to the radiologist. When a non-complex fracture or dislocation (a so-called 'minor trauma') is diagnosed, the for this purpose well-trained general practitioners provide the patient with the usual care (eg, a splint or sling) and provide follow-up consults in their practice. This so-called substitution of care from the secondary to the primary care setting is stimulated by governments and insurers in

the Netherlands.[2–5] However, while minor trauma care is provided in several general practices in the Netherlands and is supported by healthcare professionals in both general practice and hospital, it is unknown what the patient satisfaction level is and which determinants affect it. This is remarkable because patient satisfaction is considered as one of the key factors of a successful organisation of care.[6]

In that light we aim to study patient satisfaction towards minor trauma care for non-complex fractures or dislocations in the primary care setting in comparison to the secondary care that is, hospital setting. When the general practitioners in our study obtain similar results as the nearby hospitals, minor trauma care may be substituted nationwide and beyond.

## OBJECTIVES

To assess patient satisfaction towards minor trauma care in the primary and secondary care setting. In addition, we aim to study demographic factors, treatment results, time consumption and costs to assess which determinants affect patient satisfaction.

## METHODS AND ANALYSIS

### Study design

This is a prospective observational cohort study including patients presenting at the X-ray facility in the general practice Zorgplein Lemmer and patients presenting at the X-ray facility of the Antonius Hospital Sneek, both located in the north of the Netherlands, with an X-ray confirmed diagnosis of a non-complex fracture or dislocation and planned to be treated in either setting.

### Hospital

The Antonius Hospital Sneek is a medium-sized hospital with 300 patient beds, almost 3000 employees and a large service area consisting of almost 150 000 inhabitants. Per year, more than 14 000 patients consult the emergency department, of which a notable part is related to minor traumas.[7] Minor trauma care (treatment of non-complex fractures and bone dislocations) is mostly provided by emergency care doctors, under supervision of (trauma) surgeons. When a radiologist diagnoses a non-complex fracture or dislocation, the emergency care doctor clinically assesses the patients and evaluates the X-ray diagnosis. When the emergency care doctor agrees with the radiological diagnosis, he composes a treatment plan. When needed, he may assess a trauma surgeon for supervision. The trauma surgeon provides follow-up consults in his outpatients' clinic. Treatment, follow-up consults, all procedures and management are provided in accordance to the standards of surgical trauma care in the Netherlands.

### General practice

Zorgplein Lemmer is a general practice where regular first-line general medical care is provided by three general practitioners, supported by nurse practitioners, nurses and doctor's assistants.[8] The Antonius Hospital Sneek has recently equipped this general practice with a regular X-ray facility, which is operated by a radiographer who is employed by the hospital. Digital images are transmitted to in the Antonius Hospital Sneek, where they are assessed by a radiologist. When a non-complex fracture or dislocation is diagnosed, the general practitioner is asked to clinically assess the patient, as well as to evaluate the X-ray diagnosis. When the general practitioner agrees with the diagnosis and no contraindications exist for treatment in the general practice (eg, severe divergent bone position, suspicion of damage to nerves, vessels or tendons), the general practitioner composes a treatment plan according to the treatment protocol.[9] The general practitioners of Zorgplein Lemmer and LemmerRijn received training in minor trauma care from the hospital surgeons. When needed, the general practitioner telephonically assesses a trauma surgeon from the Antonius Hospital Sneek, who is able to assess the X-ray as well. This general practitioner also provides follow-up consults in his practice. Treatment, follow-up consults and all procedures are provided similar to the hospital's standard of trauma care, which are equal to the standard of surgical trauma care in the Netherlands.

Any treatment, which may not be specifically described in this manuscript, study protocol or treatment protocol, is provided according to the standard of surgical care in the Antonius Hospital Sneek and national guidelines.

### Participants

For participation in this study, eligible patients must meet these inclusion criteria:
1. X-ray confirmed diagnosis of a non-complex fracture or dislocation, which can be treated in the primary care setting according to the treatment protocol.[9]
2. Ability of the patient to comprehend the provided patient letter, information brochure and informed consent form.
3. A signed and dated written informed consent form. Parents of patients of age 12–17 must provide a signed and dated written informed consent form as well.

Exclusion criteria:
1. Patients aged 11 years and younger.
2. Patients presenting outside office hours, that is, Monday to Friday, 08:00–17:00 hours.

### Procedures

#### Recruitment

Participating general practitioners near Lemmer will perform the assessment of eligibility. They are asked to approach each potential participant and enquire about their interest and eligibility in participation in our study. Both the Zorgplein Lemmer as well as the Antonius Hospital Sneek have been informed about the importance

of recruiting participants, by e-mail, newsletters, training sessions and presentations. When a patient agrees to participate in our study, a staff member or a researcher will go through the informed consent process, including an explanation of the purpose of the study, procedures, risk and benefits, possible alternatives to participation, and data collection, archiving, and protection. Each patient who chooses to participate will sign and date the informed consent form. Parents of participants of age 12–17 years at the date of informed consent must provide a signed and dated written informed consent as well. A photocopy of the signed and dated informed consent form(s) will be stored in the participant's medical record at the study site as well as the investigator's site file and one photocopy will be given to the participant. All participants with written informed consent will be provided with a unique study number. Both the date of providing informed consent as well as recruitment information and participant's contact information are entered into the online study database. Following the informed consent procedure, all patients who start their treatment within the study are considered as enrolled. All participants will be followed-up within the study protocol, except if their participation in the study is prematurely ended, for example, by withdrawal of informed consent. All patients recruited in the Zorgplein Lemmer or Antonius Hospital Sneek are allocated to the corresponding analysis group, respectively. This allocation scheme fits to the intention to treat approach in the statistical analysis.

### Baseline assessment

All enrolled patients will be entered into the patient electronic enrolment log identically performed at both study sites. At baseline, demographical data will be assessed, as well as details relative to the injury (impact of the trauma, side affected, fracture classification if available), and comorbidities.

### Interventions

In this study, all treatments and follow-up visits in either the Zorgplein Lemmer or Antonius Hospital Sneek will be performed in accordance to the above-mentioned hospital's standard of care.[9] The study-related questionnaires will be completed 1 week after treatment as well as 6 weeks and 12 weeks after treatment. Table 1 summarises all questionnaires as well as their time-points.

### Outcome measures

#### Primary outcome measure

Patient satisfaction measured using the Patient Satisfaction Questionnaire Short Form (PSQ-18; 12 weeks after treatment).

#### Secondary outcome measures

1. Patient satisfaction measured using the PSQ-18 (1 and 6 weeks after treatment).
2. Complications of treatment and pain scores (12 weeks after treatment).
3. Physical functioning according to the 12-item WHO Disability Assessment Schedule II (WHO-DAS 2.0; 12 weeks after treatment).
4. Limitations in functions of upper extremities (if applicable) according to the disabilities of the arm, shoulder and hand (DASH) questionnaire (12 weeks after treatment).
5. General health status according to the 12-item General Health Questionnaire (GHQ-12; 12 weeks after treatment).

**Table 1** Overview of the outcome measures and time points of assessment

| Assessment parameters | Pre-treatment | 1 Week after treatment | 6 Weeks after treatment | 12 Weeks after treatment |
|---|---|---|---|---|
| Patient information/consent | X | | | |
| Eligibility | X | | | |
| Demographics | X | | | |
| Details of injury | X | | | |
| Comorbidities | X | | | |
| Patient satisfaction: PSQ-18 | | X | X | X |
| Complications of treatment and pain scores | | | | X |
| Physical functioning: WHO-DAS 2.0 | | | | X |
| Limitations in functions of upper extremities: DASH* | | | | X |
| General health: GHQ-12 | | | | X |
| Quality of life: EQ5D | | | | X |
| Time consumption | | X | X | X |
| Costs | | | | X |

DASH, disabilities of the arm, shoulder and hand questionnaire; EQ5D, EuroQoL5; GHQ-12, 12-item General Health Questionnaire; PSQ-18, Patient Satisfaction Questionnaire Short Form; WHO-DAS 2.0, WHO Disability Schedule II.
*Only assessed in patients with a treatment of a fracture or dislocation in an upper extremity.

6. Quality of life using the EuroQoL5 (EQ5D) questionnaire (12 weeks after treatment).
7. Time consumption (waiting time, treatment time, travelling time and distance; 1, 6 and 12 weeks after treatment).
8. Costs (12 weeks after treatment).

## Instruments

1. PSQ-18 is a questionnaire to assess patient's satisfaction with healthcare.[10] This questionnaire was developed and abbreviated from larger questionnaires,[11 12] maintaining internal consistency and reliability.[10–12] Seven domains of patient satisfaction are researched with Likert scales: general satisfaction, technical quality, interpersonal manner, communication, financial aspects, time spent with doctor, and accessibility and convenience. Each dimension is tested through different questions, which is of substantial benefit when one aims to identify a particular area to improve on. Certainly, general satisfaction has strong correlation with the other domains and thus it is important to assess all different domains.
2. Complications of treatment will be assessed using an open question 'did you experience any complications of the treatment, or did you need to be operated?'. Pain scores will be examined using three Visual Analogue Scores (VAS) for (1) pain in rest, (2) pain during daily routines at home and (3) pain during activities at work. The VAS is a widely used one-dimensional measure of pain intensity.[13] The pain VAS is a continuous scale comprised of a horizontal line, anchored by two verbal descriptors, one for each symptom extreme (no pain vs unbearable pain).[14 15]
3. Physical functioning is assessed using the 12-item WHO-DAS 2.0.[16] This questionnaire was developed to evaluate patients' functioning according to the International Classification of Functioning, Disability and Health (ICF). The ICF is an integrative biopsychosocial model for comprehensively evaluating the functioning and (dis)abilities of patients. The ICF provides information on health conditions, impairments of body functions or structures, activity limitations, participation restrictions and relevant environmental effects.[17] To quantify the multidimensional aspects of patients' disability status, WHO-DAS 2.0 was developed in accordance with the ICF framework for evaluating six domains of functioning, including social participation and cognition-related daily activities. WHO-DAS 2.0 can evaluate patients' disability and functional status with adequate reliability and validity.[18]
4. If the treated fracture or dislocation is located in the upper extremities, the DASH questionnaire will be used to assess its functionality.[19] The DASH questionnaire is a 30-item, self-administered assessment of upper-extremity symptoms and disability, with a focus on physical function. A high DASH score indicates severe disability.[19]

5. The participants' general health status is assessed using the widely used GHQ-12.[20] This self-administered short-form is designed to evaluate (mental) health of study participants in a broad sense. Answers are to be given in reference to the last few weeks. The GHQ-12 comprises 12 questions regarding the general level of happiness, the experience of depressive and anxiety symptoms, perceived stress and sleep disturbance. Items are scored using values of 0, 0, 1, 1 for the answers. A decrease in the scores represents improvement.[21]
6. Quality of life is investigated using the EQ5D questionnaire, which is a general measurement of health-related quality of life.[22] The EQ5D questionnaire has gained widespread acceptance and consists of a short survey of five domain-specific questions and a VAS that takes less than 2 min to complete and has been found to be both reliable and valid.[22]
7. Participant's time consumption is assessed using a questionnaire which quantifies the waiting time (in the waiting room at the General Practitioner's office, in the waiting room of the X-ray facility, in the waiting room of the treatment facility), treatment time (at the GP's office, in the X-ray facility and in the treatment facility, travelling time and distance (from home to the GP's office, from the GP's office to the X-ray facility, from the X-ray facility to the treatment facility and from the treatment facility back to home). Time is measured in minutes and distance is measured in kilometres. Time consumption is measured at the day of treatment as well as at days of follow-up consultations. The questionnaire therefore is administered 1, 6 and 12 weeks after treatment.
8. In the Netherlands, costs of diagnostics, treatment and follow-up are defrayed by the health insurance companies. These health insurance companies will evaluate the costs in both treatment arms. Only the policy excess of €385 maximum may be charged. This policy excess is assessed in one question administered 12 weeks after treatment.

## Sample size estimation

We intended to perform the sample size calculation based on the difference in mean patient satisfaction between both groups. However, there was no literature available concerning patient satisfaction in trauma care in general practices or hospitals, let alone effect sizes. Therefore, we based our sample size calculation on feasibility. With a 5% two-sided significance level, power of 80% and two equal-sized treatment groups, a sample size of 200 participants (100 in both groups) was determined to be feasible and sufficient to demonstrate effect sizes of 0.4 (small to medium) or over.

## Statistical analyses

Our statistical analyses will be performed using an intention-to-treat approach using data from all enrolled patients and according to their initial treatment setting.

First, univariable statistical tests (ie, $\chi^2$ tests or Fisher's exact tests for categorical variables; t-test or Wilcoxon rank-sum test for continuous variables) will be performed to assess differences in outcome scores between both treatment groups which are potential confounders of the setting patient outcome relationship.

As a primary analysis, mean patient satisfaction at 12 weeks after treatment will be compared between the two settings and differences will be supplied with the 95% CIs. In addition, multivariable regression models will be used with patient satisfaction at 12 weeks as the dependent variable and treatment as well as potential confounders (eg, age, gender) as independent variables. If substantial confounding appears present the results from these models will be deemed final.

Subsequently, secondary analyses will be conducted using multivariable regression models to estimate associations of mean patient satisfaction scores with other potential determinants (eg, complications, pain scores, physical functioning, EQ-5D, time consumption as independent variables). Also, in these analyses potential confounding will be addressed. In addition, we will assess interaction between treatment and these determinants by including the pertaining product terms treatment×determinant as independent variables in the multivariable regression model and testing their statistical significance.

The cost-effectiveness of the treatments will be researched using an incremental cost-effectiveness ratio, which will be assessed by calculating differences in mean costs, divided by differences in mean Quality Adjusted Life Years (QALY)s between both treatment sites.

Data of participants who withdrew from our study follow-up for any reason (eg, withdrawal of consent, death, loss to follow-up) will be included in the analysis until the time at which the participants withdrew.

Complete case analysis can give biased results because non-response is commonly non-random. Furthermore, the exclusion of patients with missing data will decrease the statistical power of the study due to a reduced number of subjects in the analyses. We will therefore account for missing data by using multiple imputation by chained equations under the assumption that the missingness mechanism is missing at random or missing completely at random. We will impute 20 (or more if the % missing data is high) datasets and data will be pooled using Rubin's rules.[23] The imputation model will include the analysis variables as well as all variables that may predict missingness of a variable. We will study the missing data mechanism of the variables by predicting 'missingness' (yes/no) of each of these variables using a multivariable logistic regression analysis.

Subsequently, results will be discussed using a small focus group consisting of patients (n=15 per group) and healthcare providers. Patients and healthcare providers will be selected at random and will be invited for one focus group session wherein both patients reported measures and patient reported experience measures will be discussed. Results of this focus group discussion will be reported separately from the cohort study results.

### Data collection and management
Collection, processing, storing and securing of research data will be performed in accordance with the General Data Protection Regulation (GDPR) rules, International Organization for Standardization 14155 guidelines and (local) laws and regulations. For this study, online electronic case report forms (e-CRFs) have been designed in REDCap.[24] Changes of these e-CRFs will be applied only following an approved amendment to this study protocol. Access to the data and the e-CRF is protected with 'two-step' security. Prior to the enrolment of the first participant, study teams at both sites received a training programme included explanations on criteria for inclusion and exclusion, study protocol, study procedures and how to use our e-CRF. Study monitoring visits will be provided as frequently as necessary to guarantee the completeness and accuracy of the data in our e-CRFs. At the end of the patient enrolment period, both sites will be provided with a close out visit and all final clarifications will be done. All source data and any other essential documents will be archived following legal requirements at both study sites. Collected study data will be archived by the study sponsor following legal requirements.

### Premature termination
Because of the nature and design of this study, no stopping rules were defined. All provided treatments and follow-up are standard of care and no additional or divergent medication, interventions, or investigational medical devices are applied in this observational study.

### Reporting of adverse events
During the study, all adverse events (AEs) are registered. All (serious) AEs will be reported to the ethical committee in accordance to local regulatory requirements.

### Ethical considerations and dissemination
Our study results will be presented at (inter)national conferences and published in peer-reviewed journals.

### Patient and public involvement
Patients and public were not involved in the design of, recruitment to or conduct of this study. However, study results will be disseminated to all study participants by sending them an (e)mail with our study results, phrased without medical jargon.

## DISCUSSION
Due to rising costs in healthcare, governments and insurers in the Netherlands aim to relocate minor trauma care from the secondary to the primary care setting.[2–5] Patient satisfaction is considered as one of the key factors of a successful organisation of care.[6] In this study, we aimed to determine the effect of minor

trauma care in a general practice on patient satisfaction compared with treatment in a hospital. We chose to use an observational study design because this design may help us to assess the effect of the complete chain of care. The choice of X-ray and treatment location (general practice or hospital) is decided by the referring general practitioner in consultation with the patient. However, a randomised controlled study design would not have resulted in this real-world data, which was our primary objective. Furthermore, the results from this observational study are particularly important for our cost-effectiveness analysis.

The primary outcome patient satisfaction is a well-defined parameter.[10–12] However, both our primary as well as our secondary outcome measures are patient-reported outcomes, which will require compliant participants. We are aware of the risk of bias as a result of patients lost to follow-up or unwilling to finish questionnaires. Important variables, which may alter study outcomes, will be controlled during the statistical analyses. Missing values will be accounted for using multiple imputation performed according to our statistical analysis plan.

Our study results are expected to provide insight in determinants of patient satisfaction in minor trauma care in the primary and secondary care setting. While governments and insurers stimulate substitution of care from the secondary to the primary care setting, insight in determinants of patient satisfaction as well as cost-effectiveness will be of increasing importance.

## CURRENT STUDY STATUS

We started patient recruitment in November 2017. The numbers of patients recruited is as follows: Zorgplein Lemmer: 47; Antonius Hospital Sneek: 24 (December 2018). Data collection will be expected to be completed (final questionnaire of the last patient) in December 2019. This manuscript has been prepared following the Strengthening the Reporting of Observational Studies in Epidemiology-checklist.

**Author affiliations**
[1]Department of General Practice, University Medical Center Groningen, University of Groningen, Groningen, The Netherlands
[2]Department of General Practice, Huisartsenpraktijk Arentsen & Groeneveld, Lemmer, The Netherlands
[3]Department of General Practice, Huisartsenpraktijk LemmerRijn, Lemmer, The Netherlands
[4]Department of Surgery, Antonius Ziekenhuis, Heelkunde Friesland Groep, Sneek, The Netherlands
[5]Department of Surgery, Tjongerschans Ziekenhuis, Heelkunde Friesland Groep, Heerenveen, The Netherlands

**Correction notice** This article has been corrected since it first published online. The open access licence type has been amended.

**Acknowledgements** The authors thank the staff members of Zorgplein Lemmer and Antonius Hospital Sneek who have contributed to this study, directly or indirectly.

**Contributors** HB: Conception and design of the study. TV and HB: Development and ethical approval of the study protocol, data collection, drafting, revision and approval of this manuscript. HA and EJB: Patient treatment (general practice), data collection, manuscript drafting and revision, approval of this manuscript. MMK and PHWL: Patient treatment (hospital), development and approval of the study protocol, data collection, revision and approval of this manuscript.

**Funding** Both the study and the publication this manuscript were funded by the Innovation Fund of De Friesland Zorgverzekeraar, the Netherlands (grant number R135).

**Competing interests** None declared.

**Patient consent for publication** Not required.

**Ethics approval** The Medical Ethics Committee from the University Medical Center Groningen reviewed this study protocol and granted exemption from ethical approval prior to patient enrolment (number METc UMCG 2017/277).

**Provenance and peer review** Not commissioned; externally peer reviewed.

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
