## [Reviewer comments · BMJ Open]

ARTICLE DETAILS

TITLE (PROVISIONAL)	Patient satisfaction in treatment of non-complex fractures and dislocations in general practice in the Netherlands: prospective cohort study protocol
AUTHORS	Verbeek, Tjitte; Arentsen, Hans; Breet, Evert; Kuipers, Machiel; Lubbert, Pieter; Burger, H

VERSION 1 – REVIEW

REVIEWER	Reviewer name: Dr. Richard Buckley Institution and Country: Foothills Medical Center, University of Calgary, Canada Competing interests: None declared
REVIEW RETURNED	12-Jul-2018

GENERAL COMMENTS	No real comments as it is a well described study except that this reviewer would not have published until the study was completed. By publishing early, the authors may bias their results.
---

REVIEWER	Reviewer name: Martin Bach Jensen Institution and Country: Center for General Practice at Aalborg University, Aalborg, Denmark Competing interests: None declared
REVIEW RETURNED	18-Nov-2018

GENERAL COMMENTS	This is an ongoing study. In a parallel cohort study it compares patient satisfaction and other aspects relating to the treatment of non-complex fractures and dislocations in two different locations: general practice in the Netherlands and a hospital setting. This is followed by a qualitative study (focus group). The focus of the study feeds into a current trend of transforming care from hospital setting to primary care. Though not a RCT the choice of methods is sound and the protocol well written. Minor comments The recruitment seem to be slower than anticipated. It is stated that "Participating general practitioners near Lemmer will perform the assessment of eligibility. They are asked to approach each potential participant and enquire about their interest and eligibility in participation in our study. Both the Zorgplein Lemmer as well as the Antonius Hospital Sneek have been informed about the importance of recruiting participants." It would be of interest to know what additional measures are used to enhance recruitment. It should be stated that the handling of data follows the GDPR rules.
--

	It would have added to the paper if it included a discussion of the risk of not acknowledging a difference in adverse events relating to the differences in procedure. Under exclusion criteria one sentence seems to end prematurely. "2. Patients presenting outside office hours, i.e. ."
--	--

VERSION 1 – AUTHOR RESPONSE

Reviewer: 1

No real comments as it is a well described study except that this reviewer would not have published until the study was completed. By publishing early, the authors may bias their results.

Thank you very much for your compliments!

Reviewer: 2

This is an ongoing study. In a parallel cohort study it compares patient satisfaction and other aspects relating to the treatment of non-complex fractures and dislocations in two different locations: general practice in the Netherlands and a hospital setting. This is followed by a qualitative study (focus group). The focus of the study feeds into a current trend of transforming care from hospital setting to primary care. Though not a RCT the choice of methods is sound and the protocol well written.

Thank you very much for your compliments!

Minor comments

The recruitment seem to be slower than anticipated. It is stated that "Participating general practitioners near Lemmer will perform the assessment of eligibility. They are asked to approach each potential participant and enquire about their interest and eligibility in participation in our study. Both the Zorgplein Lemmer as well as the Antonius Hospital Sneek have been informed about the importance of recruiting participants." It would be of interest to know what additional measures are used to enhance recruitment.

In accordance with the reviewer comment, we added the additional measures which were used to enhance recruitment, namely by e-mail, newsletters, training sessions, and presentations. Page 9 (Methods section).

It should be stated that the handling of data follows the GDPR rules.

In accordance with the reviewer comment, we added that the handling of data follows the GDPR rules. Page 16 (Methods section).

It would have added to the paper if it included a discussion of the risk of not acknowledging a difference in adverse events relating to the differences in procedure.

While we do not address a difference adverse events as an outcome measure, all adverse events are registered during the study. All (serious) AE's will be reported to the ethical committee in accordance to local regulatory requirements. We stated this on page 17 (Methods section).

Under exclusion criteria one sentence seems to end prematurely.

"2. Patients presenting outside office hours, i.e. ."

In accordance with the reviewer comment, we added the correct office hours. Page 9 (Methods section).

Finally, we added a 'Patient and Public Involvement' statement, according to the provided instructions. Page 18 (Methods section).

We hope that you will find the revised manuscript suitable for publication.

VERSION 2 – REVIEW

REVIEWER	Reviewer name: Martin Bach Jensen Institution and Country: Center for General Practice at Aalborg University, Denmark Competing interests: None declared
REVIEW RETURNED	16-Dec-2018

GENERAL COMMENTS	The issues raised in the previous review have been addressed in a satisfactory manner.
--